# Niqivut Silalu Asijjipalliajuq: Building a Community-Led Food Sovereignty and Climate Change Research Program in Nunavut, Canada

**DOI:** 10.3390/nu14081572

**Published:** 2022-04-10

**Authors:** Amy Caughey, Pitsiula Kilabuk, Igah Sanguya, Michelle Doucette, Martha Jaw, Jean Allen, Lily Maniapik, Theresa Koonoo, Wanda Joy, Jamal Shirley, Jan M. Sargeant, Helle Møller, Sherilee L. Harper

**Affiliations:** 1Department of Population Medicine, University of Guelph, Guelph, ON N1G 2W1, Canada; sargeanj@uoguelph.ca (J.M.S.); sherilee.harper@ualberta.ca (S.L.H.); 2Department of Health, Government of Nunavut, Pangnirtung, NU X0A 0H0, Canada; pkilabuk1@gov.nu.ca (P.K.); isanguya@gov.nu.ca (I.S.); mdoucette@gov.nu.ca (M.D.); mjaw@gov.nu.ca (M.J.); tkoonoo@gov.nu.ca (T.K.); wjoy@gov.nu.ca (W.J.); 3Ilisaqsivik, Clyde River, NU X0A 0E0, Canada; 4Nunavut Tunngavik Incorporated, Iqaluit, NU X0A 0H0, Canada; jallen@tunngavik.com; 5Ilitasiniq Nunavut Literacy Council, Iqaluit, NU X0A 0H0, Canada; lilymaniapik@nunavutliteracy.ca; 6Nunavut Research Institute, Iqaluit, NU X0A 0H0, Canada; jamal.shirley@arcticcollege.ca; 7Department of Health Sciences, Lakehead University, Thunder Bay, ON P7B 5E1, Canada; hmoeller@lakeheadu.ca; 8School of Public Health, University of Alberta, Edmonton, AB T6G 2R3, Canada

**Keywords:** Indigenous knowledge, Inuit knowledge, country food, food security, food sovereignty, climate change, Indigenous methods, Indigenous health, storytelling

## Abstract

The history of health research in Inuit communities in Canada recounts unethical and colonizing research practices. Recent decades have witnessed profound changes that have advanced ethical and community-driven research, yet much work remains. Inuit have called for research reform in Inuit Nunangat, most recently creating the National Inuit Strategy on Research (NISR) as a framework to support this work. The present study details the process undertaken to create a research program guided by the NISR to address food security, nutrition, and climate change in Inuit Nunangat. Four main elements were identified as critical to supporting the development of a meaningful and authentic community-led program of research: developing Inuit-identified research questions that are relevant and important to Inuit communities; identifying Inuit expertise to answer these questions; re-envisioning and innovating research methodologies that are meaningful to Inuit and reflect Inuit knowledge and societal values; and identifying approaches to mobilizing knowledge that can be applied to support food security and climate change adaptation. We also identify considerations for funding agencies to support the meaningful development of Inuit-led research proposals, including aligning funding with community priorities, reconsidering who the researchers are, and investing in community infrastructure. Our critical reflection on the research program development process provides insight into community-led research that can support Inuit self-determination in research, enhance local ethical conduct of research, privilege Inuit knowledge systems, and align Inuit-identified research priorities with research funding opportunities in health research. While we focus on Inuit-led research in Nunavut, Canada, these insights may be of interest more broadly to Indigenous health research.

## 1. Introduction

The history of research involving Inuit communities in Canada has been described by Inuit as one of exploitation and racism, rooted in colonization and often resulting in dehumanization of Inuit [1]. Striking examples of unethical health research inflicted on Inuit in the past 70 years include post-war medical experiments by scientists wishing to learn more about cold weather exposure and “problem of human adaptability” in the context of environmental extremes [2,3], and nutrition-related human experimentation [4]. More recently, research has undergone profound changes to advance ethical research practice involving Inuit communities, yet work remains to address ongoing challenges, including ensuring that research funding priorities match community priorities, that research is of benefit to communities, and that appropriate research methods and results sharing and knowledge mobilization occurs within Inuit communities [5,6]. Inuit in Canada have called for research reform in Inuit Nunangat (Inuit homeland) [1]. In 2018, Inuit Tapiriit Kamatami (ITK) released the National Inuit Strategy on Research (NISR) which voiced the need to advance Inuit self-determination in research, identify research priorities that ensure Inuit communities are benefiting from research that is useful, that Inuit are setting research priorities, that new knowledge is meaningful and empowers communities to meet self-determined needs, and that the research community and Inuit work together to replace colonial approaches to research [1]. The NISR built on work over the past two decades by ITK and others that has outlined guidelines for how research should be conducted in Inuit Nunangat [7,8] and that has provided perspectives and clear actions that support ethical research for Inuit communities [9].

The focus on this paper is research in the context of nutrition, food security, and climate change in one region of Inuit Nunangat. Inuit in Canada have experienced a rapid nutrition transition, characterized by a decreased intake of nutrient-rich traditional, local food (“country food”) and a parallel increase in energy dense, yet often nutrient poor, market food [10,11,12,13]. The causes of this dietary transition are complex and are fueled by colonization, poverty, changing food preference and knowledge, socio-economic factors, and climate change [14,15,16,17]. Furthermore, even though it is consumed less frequently, country food remains a critical contributor to nutrient intake and health for Inuit; regular consumers of country food have higher intakes of key nutrients needed for good health, including protein, n-3 polyunsaturated fatty acids, iron, selenium, and a range of vitamins (including A, B_6_, B_12_, C, D, E) and lower intakes of carbohydrates, saturated fat, and sodium [14].

The goal of this paper is to share insights from the Inuit-led development of a research program that aims to address food security, nutrition, and climate change in Nunavut, Canada. Specifically, this article discusses: (1) the theory and placement of research that is relevant to each Inuit community; (2) the identification of research methodologies and methods that align with Inuit worldviews; and (3) the application of research knowledge to advance food security in a changing climate. We also reflect on the proposal development process and offer suggestions that could enable funding agencies to support and enhance Inuit self-determination in research. Our approach to developing an Inuit-led research program offers an example of how the NISR can be mobilized in the context of community health research to support decolonizing research practices, Inuit self-determination in research, and climate change adaptation in Nunavut. While we critically reflect on our process in one Inuit region of Inuit Nunangat, our insights may be useful to advancing the decolonization of Indigenous health and climate research more broadly. 

## 2. Materials and Methods

### 2.1. Placing of the Research

Over 65,000 Inuit live in Canada, approximately 73% of whom live in Inuit Nunangat, or “Inuit homeland”; Inuit Nunangat includes four distinct regions (Nunavut, Nunavik, Nunatsiavut, Inuvialuit Settlement Region) that occupy close to 35% of the landmass of Canada and 50% of the Canadian coastline [18]. Disparities exist for Inuit in Canada. There is a ten year gap in life expectancy between Inuit and other Canadians, with Inuit experiencing a life expectancy of 74.2 years, compared with a 84.9 year life expectancy for non-Indigenous Canadians [18]. The Inuit population in Canada is young and growing, and rates of infant mortality and pre-term birth remain higher for Inuit that for non-Indigenous Canadians [18]. While social determinants of health such as housing, food security, and access to healthcare remain challenges for many Inuit communities, Inuit representative organizations also have identified unique strength-based indicators that support health and well-being, including a strong sense of belonging to the local community (81% of Inuit vs 65% of Canadians generally) [19], a deep respect for Elders and the sharing of Inuit knowledge from generation to generation, and a high percentage (70%) of adults who participate in harvesting activities [18].

### 2.2. Methodology and Approach

This research was based on principles of community-based participatory research that privilege relationships within research teams, mutual learning, and research that responds to community identified priorities [5,20]. Such approaches are often aligned with Indigenous community-identified ethical research practices [20], including Inuit organizational guidance for ethical research practice [7]. This research was guided by decolonizing approaches to research with Indigenous Peoples [21,22], and centered on Inuit epistemologies [23], Inuit methodologies and consensus models [24,25], and prioritizing Inuit Qaujimajatuqangit (IQ)—“what Inuit have always known to be true” [26].

### 2.3. Data Collection and Analysis

There is growing discussion on the impact of climate change in the Arctic and calls for climate change research to be conducted within the socio-ecological contexts and needs of present-day Inuit communities [27] by addressing issues such as food security and health [28,29,30,31,32,33,34,35]. Food security, and the importance of Inuit country food (i.e., food that is locally hunted, trapped, fished, gathered, or otherwise harvested) in achieving food security is not only a pressing climate change challenge and health-related climate change indicator [36], but country food is also a critical contributor to Inuit health and well-being across the lifespan [12,14,37,38,39,40,41,42,43,44,45] and is the preferred food choice of many Inuit in Nunavut [15,46].

In July 2018, the Canadian Institutes of Health Research (CIHR) Institute of Nutrition, Metabolism and Diabetes issued a call for proposals for research teams to address climate change and food security in Northern Canada [47]. The objectives of this funding centered around creating new knowledge regarding the magnitude and health effects of climate change, identifying approaches to address food insecurity through implementation science, and building capacity for multidisciplinary research [48]. CIHR encouraged community-led proposals that supported community-identified research questions. Around the same time, ITK introduced the NISR, outlining expectations of how research should be conducted in Inuit Nunangat [1], as well as the National Inuit Climate Change Strategy which highlights climate change priorities related to Inuit health, environmental health and well-being outcomes, and food security for Inuit communities [49].

Guided by these strategies, during Fall 2018, a group of Nunavummiut (Inuit and non-Inuit) came together to discuss the possibility of submitting a letter of intent to the CIHR team research grant funding call. The research team was initiated by a nutritionist and a climate change practitioner, who shared the funding call with community health workers, other health professionals, and several community organizations to explore interest in discussing a possible research project. Our group represented a diverse range of disciplines, backgrounds, and worldviews: health workers (community health representatives, nutritionists, environmental health specialists, a public health physician, an epidemiologist), climate change and environment specialists, Inuit land claim organization representatives, literacy organization representatives, Elder society members, community wellness organization representatives, community-based research organizations, and one university-based researcher representing the fields of public health, epidemiology, and climate change. Most of the participants based in Nunavut had existing working relationships within the small, overlapping circles of public health work and health research; the one university-based researcher who was invited to participate had existing working relationships with some members of the team (having worked for several years with some team members) but was a new face to other members of the team. The group that came together employed a consensus model of decision making to reduce potential power imbalances.

Over the span of six months (October 2018 to March 2019), a group of approximately 20 people contributed to three group face-to-face workshops and numerous individual in-person and telephone discussions to collectively co-produce and co-create a letter of intent and a research proposal. All in-person workshops were held in Nunavut; two workshops were held in Iqaluit, and one was held in Clyde River. The resulting qualitative data were analyzed in a multilayer process, weaving together workshop notes, team member reflections, and direct quotes from participants to support the realities and complexities that exist in real life and to reflect accepted qualitative research practice [50]. 

The proposal for the “Niqivut Silalu Asijjipalliajuq (NSAP): Our Food and Climate Change” (NSAP research program) was awarded funding in 2019. The overall aim of the NSAP research program was to support research about the climate-food-health connection by (1) advancing country food knowledge; (2) pioneering Inuit-led research methods, (3) shaping policy and informing community practice, and (4) developing climate–food–health research capacity in Nunavut. The experience of building this research program revealed considerations related to Inuit leadership in research proposal development, Inuit-directed research priority setting, Inuit-identified uses of research, and decolonizing approaches to research in Inuit communities.

## 3. Results

The results presented here outline broad steps taken during the development of a research program that focused on Inuit knowledge as central to supporting country food preparation, preservation, and use for food and medicine to support climate change adaptation. Through this lens, we share insights into our research program development process in four areas: (1) Inuit-defined research questions; (2) Inuit-guided content experts; (2) Inuit-centered research methodologies and methods; and (3) Inuit-prioritized knowledge mobilization for “usable” research (Figure 1). We also offer feedback to funding organization based on our work in developing this research program.

### 3.1. “For Me, Food Security Is Having Country Food”: Inuit-Defined Research Questions

As part of decolonizing research, the NISR calls for research funding to reflect Inuit research priorities and for Inuit engagement in research design, rather than merely being consulted on priorities of outside researchers [1]. Indigenous scholar Linda Tuhiwai Smith (2012) notes that research has too often been concerned with defining legitimate knowledge and that it is therefore a significant part of the colonization process; to address this, Smith [21] suggests questioning “who defined the research problem?” and “for whom is this study worthy and relevant, and who says so?” Similar questions have been suggested to researchers by Inuit organizations and northern research institutions [7], as well as calls for involvement of community members and leaders at the outset of research projects—an approach that “engages local experts as intellectual partners, rather than as sources of information that will be analyzed by someone else, somewhere else” [27].

Inuit epistemologies (or Inuit ways of knowing) have been described by researchers and by Inuit communities [2,23,24,25,26]. McGrath calls the practice of Inuit-centered epistemology “’Inuktut knowledge renewal’—knowledge in practice, in the Inuk way” [23], recognizing that important knowledge already exists within Inuit communities. In contrast, the frequent focus of Western health-focused research (and indeed encapsulated in the motto of the organization to which we were applying to fund our research) is the “creation of new knowledge” [51]. In privileging Inuit-centered methodologies and epistemologies, the NSAP research program built on sharing stories that already exist within communities and focused on renewing Inuit knowledge and the “ways of being and looking at things that are timeless” [26] to advance nutrition, food security, and climate change adaptation.

The consensus within our team, and the underpinning of our research approach, was the understanding that Inuit knowledge is knowledge that is critical to food security and climate change in Nunavut communities. Our group saw Inuit *Qaujimajatugangiit* (IQ) as encompassing a broad definition, literally translated to be “Inuit way of doing things: the past, present, and future knowledge, experience and values of Inuit Society” [46], and “ways of knowing and being” [23]. IQ has been proposed as “an ethical framework and detailed plan for having a good life… connecting all aspects of life in a coherent way” [26], which we believed could effectively ground our proposed research, respecting the need for “checkpoints” as the research moves forward to ensure the work remains connected to IQ.

A key factor in developing the research program was being able to spend time as a group defining concepts such as climate change, nutrition, and food security; in doing so, our team applied the Inuit research methodological principles of *naalangniq* (listening), *pittiarniq* (accuracy), *ujjiqsuiniq* (observation), and *suliniq* (personal congruence) [23]. While the funders had their own definitions and understandings of these concepts, the act of Inuit (re)defining key concepts related to climate change, nutrition, and food security grounded our group in the themes and proposed activities that would be central to our proposal and our research program. For instance, we discussed at length the meaning of food security, which included concepts of country food sharing, the knowledge of how to prepare and eat all parts of an animal, the identification of foods that are meant for children, women, and men, and the state of happiness related to eating country food.

The emphasis on Inuit-led solutions to food insecurity and the grounding of food within self-determination and Inuit rights to food could better be described as “food sovereignty” [46,52,53,54]. The detailed knowledge related to country food preparation and preservation, and the language associated with these practices, was identified by our group as knowledge that is being lost as Elders pass on. This concern is shared among other communities in the circumpolar north [55].

The process of redefining terms (such as food security, climate change) and establishing Inuit knowledge as core to the research program were important early steps in identifying research questions and who was best placed to answer these questions. Research that collaborates with community members at the outset has been identified as crucial in climate change to ensure the information and knowledge produced is usable and applicable for communities [27]; moreover, we believe this is an example of moving “from exclusion to self-determination in research”, as advocated for in the NISR [1].

### 3.2. “There Are People in Our Community Who Know about These Things”: Inuit-Guided Definition of Who Holds Expertise

The urgency of preserving country food knowledge became a central and important theme to the research program development. Within our group, we also asked ourselves questions such as “who is the researcher?” and “who and what should be researched?” For instance, in the sphere of research, the term highly qualified personnel (HQP) often refers to graduate students and university professors. Our group recognized that this narrow definition excluded many of the people our team had identified as holding critical knowledge related to country food and climate change, including Inuit Elders, hunters, community health workers, and other community members. Elders and other community knowledge holders with expertise in preparation of country food were identified in our research program as HQP with unique and deep knowledge, who could speak about safe food handling practices and preservation methods of country food and country food use as medicine, as well as the unique language associated with these. In redefining HQP, we highlighted Inuit knowledge systems and Inuit in Nunavut as experts with critical contributions to understanding food–health–climate relationships.

### 3.3. “The Stories Are the Research”: Inuit-Centered (Re)defining Methodologies

Around the world, Indigenous methodologies are challenging Western research paradigms and opening up opportunities for Indigenous self-determination in research [56]. Research methodology, notes Smith, is “based on the skill of matching the problem with an ‘appropriate’ set of investigative strategies… to guarantee validity and reliability”; this requires having an understanding of the problem and the method” [21], which is best understood by the Indigenous community involved in the research.

Indeed, there is a need for Indigenous health research to reflect local perspectives and Indigenous ways of knowing in health research design and data collection [20]. Inuit knowledge has been identified as critical to inform understandings in Inuit nutrition and food security research [14,15,43]. Storytelling is often central to Indigenous methodologies [22]. Sharing stories is grounded in Inuit epistemology and reflective of Inuit societal values, including *unikkaaqatiginniq*, which relates to the power of story and story-telling in concepts of well-being [24]. Storytelling has been identified as a decolonizing research method and is often considered to be a culturally appropriate research tool for representing the “diversities of truth”, where the storyteller retains control, and not the researcher [21]; further, the connectedness that is inherent to storytelling can function to engage community members in research and ensure that research participants are respected as equal partners in the research [57].

Qualitative methodologies, such as storytelling, are increasingly recognized in the health sciences as valid and essential methods of enquiry [50,58]. Research incorporating local understanding and knowledge sharing has been called for in health research; Indigenous scholar Smylie and colleagues propose that a major contributor to the “ineffectiveness of public health programs in Indigenous communities is that externally imposed strategies fail to consider local understandings of health and illness and local mechanisms of sharing knowledge” [59]. Research centered around local understanding and perspectives exists, and our team drew inspiration from work within Inuit Nunangat that was Inuit-led and that produced usable knowledge for communities [5,25,30,60,61].

The development of the NSAP research program was premised on community-led, participatory research philosophies to facilitate a decolonizing approach to knowledge mobilization [21,31,57,62]. The team that came together to develop the proposal shared interest and vision in mobilizing Inuit knowledge already held in Nunavut to support country food sovereignty; how we would carry out this vision was mapped out over multiple meetings and discussions.

In an early team meeting, we began with the question “where do we start?” to address country food security in a changing climate, and the response was clear from Inuit around our table: we must start with stories and learn from Elders and community knowledge holders who had lived experience and an understanding of country food handling, preparation, and preservation. A pivotal moment in our discussions was the expression of a team member who stated, “the stories are the research”, which was echoed by numerous team members who stated that “stories are everything” and “stories are how we (Inuit) learn and understand”.

From this, we discussed how stories and storytelling related to country food preservation and country food use could be a central component of our research proposal, and ultimately of understanding the climate-food-health connection. Our team identified stories as important in supporting food skills needed for preparing and preserving country food and for sharing knowledge related to healthy foods consumed in the past. Equally important was the language of country food and the desire to preserve Inuktitut terms related to these various processes. The concept of storytelling was woven through our proposal development: stories from Elders of preparing, preserving, sharing, and eating all parts of an animal; stories describing country food use as medicine; stories of how country food preparation methods have changed over time; and stories that can support keeping country food safe in a changing climate.

Stories serve two main functions in the NSAP research program: inform food and climate research understanding (and be shared at academic conferences and in research papers such as this one), and most importantly, enhance knowledge sharing among community members to support country food availability and healthy eating (stories documented by video/audio that can be shared in community-based programs such as prenatal nutrition programs, at health centres, in schools, and with other communities across Nunavut and beyond).

### 3.4. “This Excites Me!”: Inuit-Prioritized Knowledge Mobilization for Usable Research

The NISR outlines that research must be useful to Inuit and Inuit communities. The question of “what is useful knowledge?” has been previously asked within Nunavut. Karetak and Tester, in describing Inuit *Qaujimajatuqangit* in the context of education systems, discuss the relevance of knowledge being related to the application of knowledge, stating that “for Inuit, knowledge without application has no value” [26].

Knowledge application in Indigenous communities has been described by Smith: “A large part of the research stories that need to be told are small stories from local communities across time and space, in other words the stories that map devastation across generations and across landscapes, or the stories of transformation and hope that can also be tracked this way” [21]. In this way, knowledge use by the community is prioritized. Indigenous knowledge historically has been—and often continues to be—marginalized by Western research [25]. Elders have worried about the “disintegration” of Inuit knowledge by Western science and have observed that Inuit knowledge has been “discounted as quaint and anecdotal… where Inuit views of the world were given credence only when restated through the perspectives of southern researchers” [2].

Knowledge application, and how this would happen, was important to our team. Central partners in the NSAP research program are community health representatives (CHRs), all of whom are Inuit with extensive community-health experience, who identified Inuit knowledge to be a priority for supporting country food safety and healthy eating through country food. All CHRs involved in developing the research program had first-hand knowledge of country food history and preparation, of foodborne illness outbreaks, and of community food security priorities. All members of the research team identified priorities in their communities and in their work related to country food access and preparation, language related to food preparation, the need to share knowledge related to country food as food and medicine, with a desire to share food histories to support health and nutrition. Importantly, the team was able to see how this knowledge could be shared in communities; resources such as slow TV and digital archives that support learning through patient observation [61], a country food atlas, and “recipe” books were identified as useful, applied research “outputs”. As one team member stated at the final proposal writing workshop, “This research project excites me! I think others in my community would be interested in this research.”

### 3.5. Considerations for Funding Agencies

The process undertaken by our team to develop the NSAP research program revealed considerations for funding organizations working in similar settings.

(1) *Align funding with community priorities:* This work supports the recommendations of the NISR and previous research in the circumpolar north calling on funding organizations to align Inuit priorities and methodologies [1], in particular those that engage with Inuit knowledge and that result in knowledge that is useful at the local level [5,63]. Methodologies that respect Indigenous oral storytelling histories exist [23,25,60], yet such qualitative-based methods are far less represented in the Inuit health literature [43] despite systematic approaches to analyzing qualitative data to advance knowledge in health research [58]. By privileging the priorities of Inuit community health and wellness workers in determining how best to support food security and nutrition, the NSAP research program will utilize storytelling to support country food use and country food safety in a changing climate, and it is an example of funding being awarded that aligns with community priorities.

(2) *Recognize “who are the researchers”*: We observed that even when funding opportunities align with community priorities, there are challenges with established research processes that favor existing academic structures which may be inaccessible to, or inappropriate for, community-based researchers. For example, the submission of a “common CV” was a barrier for some of our team being named as co-investigators, despite their expertise in the foundations of our research program (country food, Inuktitut language, and storytelling methodologies). Moreover, some team members preferred to be a part of the “collective” teamwork, instead of highlighting individual accomplishments and accolades, as is the purpose of a CV. Inuit societal values emphasize non-hierarchal, collective actions in knowledge sharing and worldview [64]; recognizing such barriers to community research leadership and supporting Inuit-identified “highly qualified personnel” may work toward addressing reconciling these significant challenges.

(3) *Invest in community-based research capacity.* The NISR outlines the need for investment in both built infrastructure and human resources to advance appropriate and ethical research in Inuit Nunangat [1]. Basic research tools such as computer and phone access, internet, office space, and financial management infrastructure can be barriers for community members to engage with research and to lead research—in particular, for those not already employed in jobs that provide such. In Nunavut, few communities operate community-based research centres to support community-led research and facilitate engagement with invited researchers [5,24,25]; however, communities with these organizations can be examples of how to advance research capacity at the community level. Consideration for supporting research engagement should not be limited to researchers flying to the north to learn from communities but could also include opportunities for building relationships and discussing research priorities across northern communities in Inuit Nunangat and with other circumpolar Inuit regions. There may also be interest for some community-based researchers to travel to southern institutions to take advantage of training opportunities or to provide education to university-based researchers on research expectations in their community.

## 4. Discussion

The development of the NSAP research program privileged Inuit knowledge in community-led health research to inform environmental health, nutrition, wellness, language, and climate change adaptation by sharing stories of country food preparation, preservation, and use for health and wellbeing. Our team was guided by the call for research that is Inuit-led from the start and that upholds Inuit self-determination, partnership, and transparency [1].

Throughout, our team was guided by decolonizing approaches to research that prioritize Indigenous centered methodologies and methods, including those that align with Inuit epistemologies and that are practiced by our team members [21,24,25,30,31,56,65]. Inuit voices were prioritized in contributions and discussions and guided the ethos and direction of the research program development. Still, while all our team except for one person lived and worked in Nunavut, approximately half of our team were non-Inuit individuals working in health, community wellness, and environment in Inuit communities. Understanding and acknowledging biases that affect assumptions and reactions was key for non-Inuit team members [6]; as well, listening was an especially important activity to support a safe space and support relationships within the team [23].

Although our team conducted three workshops to develop the research program, more time and funding could have allowed workshops in additional communities to help to develop the research program. Nevertheless, the group used available funds to bring team members together in Nunavut to develop a genuine community-led research program from the very start of research program discussions and engaged local expertise as intellectual partners to ensure that the research is useful and applicable in communities [27]. Our team included Inuit community leaders who facilitated the development of research questions that were important to communities and formed the core of the proposed research program. At the same time, the involvement of a university-based researcher and others familiar with the lengthy and difficult application process was key to our success in being awarded funding. Researchers who are skilled and able to lead the synthesis of ideas and priorities to create a research proposal, compile the proposal in the format that was acceptable to the funder, support comprehensive literature reviews, and assist with tasks such as uploading CVs with good-quality university internet access were critical in the process of submitting our proposal.

Finally, of paramount importance to this work was that our team consisted of relationships and partnerships that were developed over many years. Existing working relationships, and friendships, supported trust while enabling meaningful, rich conversations and idea sharing informed by years of conversations on the topics of health, country food, tradition, and the food–environment–health connection. Because of this established trust, the team was able to draw on previous idea sharing to determine what would be good research questions; our group held a shared understanding of the centrality of country food for Inuit in Nunavut and a shared vision of prioritizing Inuit voices and grounding research within the context of Inuit-driven methodologies.

Overall, the NSAP research program aspires to advance Inuit food sovereignty for climate change adaptation and to share Inuit knowledge to support community nutrition, health, and well-being. The global COVID-19 pandemic has presented a significant challenge in moving our research forward, given that many participants involved in the research have been involved in COVID response in Nunavut and that in-person contact has been severely limited in Nunavut. Our team continues to plan for moving this research forward and to maintain the urgency of addressing country food sovereignty for Inuit communities.

## 5. Conclusions

The NSAP research program development privileged Inuit knowledge and Inuit *Qaujimajatuqangit* in research to support food security, food safety, and food sovereignty and to mitigate negative health effects of climate change. Furthermore, the funding of the NSAP research program by the Canadian Institutes of Health Research Institute of Nutrition, Metabolism and Diabetes highlights a positive example of recognition of Indigenous methodologies, prioritizing of Inuit community-based research priorities, and valuing Inuit knowledge renewal and knowledge sharing in health research. We hope this paper offers useful insights for researchers, and funders, who are interested in advancing research that aims to serve the self-determined needs of Indigenous communities.

## Figures and Tables

**Figure 1 nutrients-14-01572-f001:**
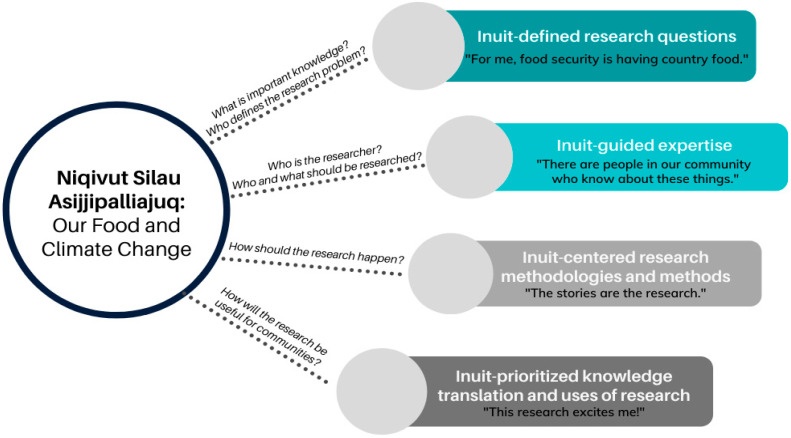
Four key elements addressed during the development of the “Niqivut Silalu Asijjipalliajuq: Our Food and Climate Change” research program.

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
