# Peer review of "Niqivut Silalu Asijjipalliajuq: Building a Community-Led Food Sovereignty and Climate Change Research Program in Nunavut, Canada"

_nutrients, 2022, doi:10.3390/nu14081572_

Round 1

Reviewer 1 Report

Welldone on your effort. This is a well-written paper.  

See below my comments:
1) Please check the spelling in lines 59, 88, 89 and across the entire manuscript.
2) Please include a literature review around your topic in the introduction section.
3) Please elaborate on the methodology of the research e.g sample size, power calculation, composition/characteristics of study participants, method of data collection, reflexivity, method of data analysis, etc. Without a proper method section this paper is incomplete.

Author Response

Peer Reviewer 1

Reviewer Comment: 1) Please check the spelling in lines 59, 88, 89 and across the entire manuscript.

  • Author's response: Thank you for this, and apologies for this not being corrected before submission. Spell check has been completed for the entire document.

Reviewer Comment: 2) Please include a literature review around your topic in the introduction section.

  • Author's response: Thank you for this feedback. Extensive edits have been made to the introduction, including additional background information and addition of references.

Reviewer Comment: 3) Please elaborate on the methodology of the research e.g sample size, power calculation, composition/characteristics of study participants, method of data collection, reflexivity, method of data analysis, etc. Without a proper method section this paper is incomplete.

  • Author's response: Thank you for this helpful feedback. The methods section has been extensively edited to elaborate on these factors, including research approach, Indigenous methodologies, methods, data analysis, as well as further elaboration on the study participants.

Reviewer 2 Report

Overall this is a well written paper. it underlines the need to assess Inuit - Nunavut, a population that needs better understanding and interventions that adhere and respect their methods, culture and well-being. 

My only recommendation is for the authors to include more nutrition related information. During my studies in Canada, I came to understand and assess "Western" errors performed regarding nutrition interventions that did not primarily examine problems in adherence and especially effectiveness. 

These lead to further problems in nutritional inadequacies and food safety. The aim was then to teach these communities their traditional ways. Can the authors please expand

Author Response

Peer Reviewer 2

Reviewer Comment: Overall this is a well written paper. it underlines the need to assess Inuit - Nunavut, a population that needs better understanding and interventions that adhere and respect their methods, culture and well-being. 

  • Author's response: Thank you for this encouraging feedback.

Reviewer Comment: My only recommendation is for the authors to include more nutrition related information. During my studies in Canada, I came to understand and assess "Western" errors performed regarding nutrition interventions that did not primarily examine problems in adherence and especially effectiveness. These lead to further problems in nutritional inadequacies and food safety. The aim was then to teach these communities their traditional ways. Can the authors please expand.

  • Author's response: Thank you for this very valuable suggestion. An additional nutrition focused paragraph has been added to the introduction, and this has been considered throughout the edits, including extensive edits to the methods section, highlighting the self-determination approach, and the prioritization of Inuit epistemologies and decolonizing methodologies in this research.